# Stanniocalcin-1 (STC-1), a downstream effector molecule in latanoprost signaling, acts independent of the FP receptor for intraocular pressure reduction

Gavin W. Roddy◉*, Tommy A. Rinkoski, Kjersten J. Monson, Uttio Roy Chowdhury, Michael P. Fautsch

Department of Ophthalmology, Mayo Clinic, Rochester, Minnesota, United States of America

* roddy.gavin@mayo.edu

**Data Availability Statement:** All relevant data are within the paper and its Supporting Information files.

## Abstract

Prostaglandin F2 alpha (PGF2α) analogues such as latanoprost are common first-line intraocular pressure (IOP) lowering medications. However, their clinical use is limited in some patient populations due to minimal or no IOP lowering response or side effects. In searching for a more targeted approach for IOP reduction, our lab recently identified Stanniocalcin-1 (STC-1) as a molecule that was required for latanoprost-mediated IOP reduction and also acted as a stand-alone IOP lowering agent. In order to determine whether latanoprost and STC-1 were equivalent and/or additive for IOP reduction, we treated C57BL/6J mice with one or a combination of these agents and measured IOP. Importance of the FP receptor for latanoprost- and STC-1-mediated IOP reduction was examined in C57BL/6J mice utilizing the pharmacologic FP receptor inhibitor AL-8810 as well as FP receptor knockout mice generated in our laboratory. Latanoprost-free acid (LFA) and STC-1 reduced IOP to a similar degree and were non-additive in C57BL/6J mice. As expected, the IOP lowering effects of LFA were abrogated by pharmacologic inhibition of the FP receptor with AL-8810 and in FP receptor knockout mice. In contrast, STC-1 maintained IOP-lowering effects in the presence of AL-8810 and also in FP receptor knockout mice. These results suggest that LFA and STC-1 show equivalent and non-additive IOP reduction in C57BL/6J mice and that unlike LFA, STC-1-mediated IOP reduction occurs independent of the FP receptor.

## Introduction

Glaucoma is the leading cause of irreversible visual impairment, estimated to afflict up to 80 million people worldwide [1, 2]. Currently, all available treatments for glaucoma are directed at reducing elevated intraocular pressure (IOP), the most prevalent and only treatable risk factor for glaucoma. Generally, eye drop therapy is initially used for IOP reduction, and of the available medications, prostaglandin F2 (PGF2α) analogues are often used as first-line IOP reducing therapy for patients with glaucoma and ocular hypertension [3]. Prostaglandins in

**Funding:** Research reported in this publication was supported by the National Eye Institute grants EY21727 (MPF) and EY26490 (MPF), Mayo Foundation (GWR, MPF), American Glaucoma Society (GWR), American Society of Cataract and Refractive Surgery (GWR), and Heed Foundation (GWR). The funders had no role in study design, data collection and analysis, decision to publish, or preparation of the manuscript.

**Competing interests:** GWR has intellectual property related to STC-I. Specific information is included below. None of these patents are related to STC-I and intraocular pressure reduction. GWR has received no royalties from any of these patents. GWR has no products in development. No other co-authors declare conflicts of interest. This does not alter our adherence to PLOS ONE policies on sharing data and materials. Title Patent Number Protein therapy for corneal inflammation, epithelial wound 8759298 healing, and photoreceptor degeneration Robert Rosa, Gavin W. Roddy, Darwin J. Prockop Adult stem/progenitor and stem cell proteins for treatment of 8785395 eye injuries and diseases. 1 Country Date Filed Date Issued US 07/2012 06/2014 US 10/2011 06/2014 Darwin J. Prockop, Joo Youn Oh, Gavin W. Roddy, Robert Rosa, Barry A. Berkowitz Adult stem/progenitor and stem cell proteins for treatment of eye injuries and diseases Darwin J. Prockop, Joo Youn Oh, Barry Berkowitz, Gavin W. Roddy, Robert Rosa 9062103 US 11/2014 06/2015 Adult stem/progenitor and stem cell proteins for treatment of eye injuries and diseases Darwin Prockop, Joo Youn Oh, Barry Berkowitz, Gavin Roddy, Robert Rosa 9730961 US 08/2015 08/2017

general are a group of pro-inflammatory molecules that signal through G-protein coupled receptors (GPCR) [3]. The particular GPCR through which the PGF2α analogues signal is known as the Prostaglandin F (FP) receptor. Binding of PGF2α analogues to the FP receptor leads to activation of a number of signaling molecules and pathways that include phosphatidy-linositol, protein kinase C, MAP kinase, and beta-catenin/T cell factor [3]. Up to 20% of patients with ocular hypertension or multiple types of glaucoma including low tension, primary open angle, exfoliative, or pigment dispersion have a reduced or absent response to PGF2α analogues [4–6]. Variability in PGF2α analogue response has been associated with single nucleotide polymorphisms (SNPs) in the FP receptor [7–12].

Though generally well-tolerated, PGF2α analogues have notable side effects such as conjunctival hyperemia, ocular surface irritation, hyper-pigmentation of the iris and periocular skin, orbital fat atrophy, and hypertrichosis [13]. PGF2α analogues have also been linked with intraocular inflammation, [14, 15] reactivation of herpes simplex keratitis, and macular edema particularly in patients with a history of uveitis or those in the post-operative period [16]. Furthermore, treatment with PGF2α analogues has been associated with an increase in markers of ocular surface inflammation in patients [17] and in animal models [18].

PGF2α analogues reduce IOP primarily by increasing outflow facility via the non-conventional or uveoscleral pathway. This occurs by matrix metalloproteinase remodeling of the extracellular matrix leading to subsequent changes in the resistance of outflow in the ciliary muscle [19–22]. While the specific effector molecules involved in PGF2α analogue activation of FP receptor-mediated IOP reduction are not completely understood, our laboratory recently identified Stanniocalcin-1 (STC-1), a multifunctional peptide hormone, as a key downstream effector molecule in this pathway. [23] In addition, STC-1 has IOP reducing activity as a stand-alone agent [23]. To further understand the relationship between latanoprost and STC-1, we sought to compare the IOP lowering properties of each drug as well as determine whether they both require the FP receptor for IOP reduction.

## Methods

### Mouse experiments

All animal studies and treatment protocols were approved by the Mayo Clinic (Rochester, MN) Institutional Animal Care and Use Committee and adhered to the ARVO Statement for the Use of Animals in Ophthalmic and Vision Research. Animals were housed with 12 hour light and dark cycles and had unrestricted access to food and water. Animals were humanely euthanized using carbon dioxide overdose at the end of the experimental period.

### Generation of FP receptor knockout mice

FP receptor knockout mice were developed in collaboration with the Transgenic and Knockout Animal Core Facility at Mayo Clinic, Rochester, MN, using the CRISPR/Cas9 strategy [24, 25]. Briefly, FP receptor knockout mice were generated by co-injection of single-guide RNA (sgRNA), WT Cas9 mRNA (L-6125, TriLink Biotechnologies, San Diego, CA) and a single-strand DNA (ssDNA) donor into the cytoplasm of C57BL/6NHsd zygotes. The ssDNA donors contained stop codons in all three frames to ensure a premature stop when inserted. An sgRNA was prepared using EnGen sgRNA Synthesis Kit, S. pyogenes (E3322S, New England Biolabs, Ipswich, MA) following manufacturer's instructions. The oligonucleotide used was Ptgfr NEBoligo 5'-TTCTAATACGACTCACTATAGTTTGGCCACCTTATCAACGGGTTTTA-GAGCTAGA-3'. The synthesized sgRNA was purified by RNA Clean & Concentrator Kit (R1017, Zymo Research, Irvine, CA) following manufacturer's instructions. The injection

mixes contained 100 ng/µl Cas9 mRNA, 50 ng/µl sgRNA and 100 ng/µl ssDNA donor in water. Surviving embryos were transferred to oviducts of pseudo-pregnant iCR mice.

Homozygous male founders were generated and crossed with C57BL/6J wild-type female mice to produce a heterozygous F1 generation. Male founders were used in crosses as FP receptor knockout females are incapable of initiating parturition. [26] Heterozygous FP receptor mice were used for inbreeding to produce F2 litters. Genotyping was performed with genomic DNA isolated from mouse tail tips using Platinum Taq polymerase (10966–018, Thermo Fisher Scientific, Waltham, MA) and primers designed to capture the insertion of the stop cassette (5' primer 1, 5'-AACACAACCTGCCAGACGGA-3'; 3' primer 2, 5'-GGAGGCA−TAGCTGTCTTTGTA-3'). For validation of FP receptor knockout success at the protein level, a quantitative sandwich ELISA was performed. Uterine tissue was selected due to high tissue expression of the FP receptor [27] and was collected post-mortem from genotyped mice and homogenized in kit dilution buffer. Total protein concentration was performed by Bradford assay. A mouse FP receptor ELISA kit (MyBioSource, San Diego, CA) was used to assay each sample in triplicate, and results were read using a TECAN Infinite M200 plate reader (TECAN, Switzerland).

## IOP experiments

IOP was measured in conscious mice using a handheld rebound tonometer (Icare TonoLab; Colonial Medical Supply, Franconia, NH). For IOP measurements, the tonometer was held perpendicular to the cornea following the manufacturer's instructions. For each time-point, three sequential but independent readings were obtained and averaged. For each independent reading, the tonometer takes six readings, discards the highest and lowest values, and shows the average of the remaining four values as a single IOP measurement. For each experiment, a second laboratory member who was masked to treatment groups checked IOP at multiple points to ensure accurate data collection.

For wild-type mouse experiments, baseline IOP measurements were recorded for two days, and then mice were randomized to one of six treatment groups (n = 6 per group): 1) latanoprost-free acid (LFA; $10^{-4}$ M; Cayman Chemical, Ann Arbor, MI) alone; 2) STC-1 (2.5 µg/µL; Biovender Research and Diagnostic Products-Czech Republic) alone; 3) FP receptor inhibitor AL-8810 (10 mM; Sigma Aldrich, St. Louis, MO) alone; 4) LFA + STC-1; 5) LFA + AL-8810; or 6) STC-1 + AL-8810. In all cases, the contralateral eye was treated with vehicle (PBS for STC-1; dilution of dimethyl sulfoxide (DMSO) 1:1000 in PBS for LFA, dilution of DMSO 1:2.5 in PBS for AL-8810). Mice were treated once daily in the morning with topical instillation of 5 µL of medication for five consecutive days followed by three days of washout in which the animals received no treatment. In cases where an animal was treated with more than one drug, the first drug was applied and the second drug was given 10 minutes later. For receptor inhibition experiments, AL-8810 was given first followed by LFA or STC-1. IOP was measured three times daily at 1, 4, and 23 hours post-treatment. The 3 IOP measurements were averaged and reported as the daily IOP. A second laboratory member who was masked to treatment groups checked IOP at multiple points during the experiment to ensure accurate data collection.

In experiments utilizing FP receptor knockout mice, IOP was measured for 4 consecutive days to obtain baseline IOP values. FP receptor knockout mice (n = 7) and C57BL/6J wild-type controls (n = 7) were treated for 6 consecutive days with topical LFA (5 µL of a $10^{-4}$ M solution) followed by cessation of treatment for 5 days. With IOP at baseline, the same mice were treated with STC-1 (5 µL of a 0.5 µg/µL solution) once daily for 6 consecutive days followed by a final 3 day washout period. IOP was measured twice daily at 1 and 23 hours post treatment.

Daily IOP measurement was reported as the average of the 1 and 23 hour IOP reading. Longitudinal data is presented as change in IOP compared to the fellow eye.

## Statistics

Average IOP was calculated at the maximal reduction of IOP with treatment. In experiments in which there were 5 treatment days, day 4 and 5 were averaged and reported as a single value. For experiments in which there were 6 treatment days, days 4–6 were averaged and reported as a single value. Given the sample size and distribution, a Wilcoxon signed rank test was used to compare control versus treatment in the same animal. In comparisons of different conditions among animals, a Kruskal-Wallis test was used with a Bonferroni correction. To determine whether there was a different response to wild-type versus FP receptor knockout mice, an analysis of variance (ANOVA) on the paired difference was used. Statistical significance was determined if $P < 0.05$.

## Results

### Comparing LFA and STC-1 treatment for IOP reduction in vivo

In order to compare STC-1 and LFA and determine whether the IOP lowering effects were additive, C57BL/6J wild-type mice were treated with either LFA, STC-1, or a combination of LFA and STC-1 given sequentially at the same concentration and amount as when applied alone. After establishing a pre-treatment baseline, STC-1, LFA, and a combination of both medications all reduced IOP compared to the fellow eye treated with vehicle, and recovered to baseline following withdrawal of the medication (Fig 1A). At treatment days 4 and 5, STC-1 ($12.7 \pm 1.0$ vs $15.3 \pm 0.9$ mmHg, 17% reduction, $P < 0.05$, n = 6), LFA ($12.8 \pm 1.3$ vs $16.1 \pm 0.8$ mmHg, 21% reduction, $P < 0.05$, n = 6) and combination therapy ($11.6 \pm 0.8$ vs $14.3 \pm 0.8$ mmHg, 19% reduction, $P < 0.05$, n = 6) all showed significant IOP reduction compared to the vehicle-treated contralateral eye (Fig 1B). The treatment groups (i.e. STC-1 vs LFA or either medication vs combination therapy) were not statistically different from one another (Fig 1B). This suggests that STC-1 and LFA have similar IOP reductions and that when combined; they do not have an additive effect.

### Determining the effect of FP receptor inhibition on IOP with LFA or STC-1 treatment

To determine whether the FP receptor was necessary for STC-1-mediated IOP reduction, C57BL/6J wild-type mice were treated with AL-8810, a pharmacologic inhibitor of the FP receptor, in combination with STC-1, LFA, or by itself. After baseline IOP measurements, only STC-1 + AL-8810 lowered IOP and subsequently returned to baseline following withdrawal of the medication. Neither LFA + AL-8810 or AL-8810 alone showed change in IOP throughout the experiment (Fig 2A). At treatment days 4–5, STC-1 + AL-8810 maintained IOP lowering capacity ($12.4 \pm 0.7$ vs $16.9 \pm 0.8$ mmHg, 22% reduction, $P < 0.05$, n = 6). However, neither LFA + AL-8810 ($15.7 \pm 0.3$ vs $15.0 \pm 0.8$, 0.05% increase, $P > 0.1$, n = 6) or AL-8810 alone ($15.1 \pm 0.6$ vs $15.3 \pm 0.6$ mmHg, 2% decrease, $P > 0.2$, n = 6) significantly affected IOP (Fig 2B).

### Generation of FP receptor knockout mice

To examine more closely the results obtained in the pharmacologic inhibition experiment above, we generated FP receptor knockout mice using CRISPR/Cas9. A Schier stop cassette was introduced within exon 2 of the *Ptfr* gene, producing a non-functional truncated protein product (Fig 3A and 3B). Genotypes of heterozygous and homozygous mice were confirmed

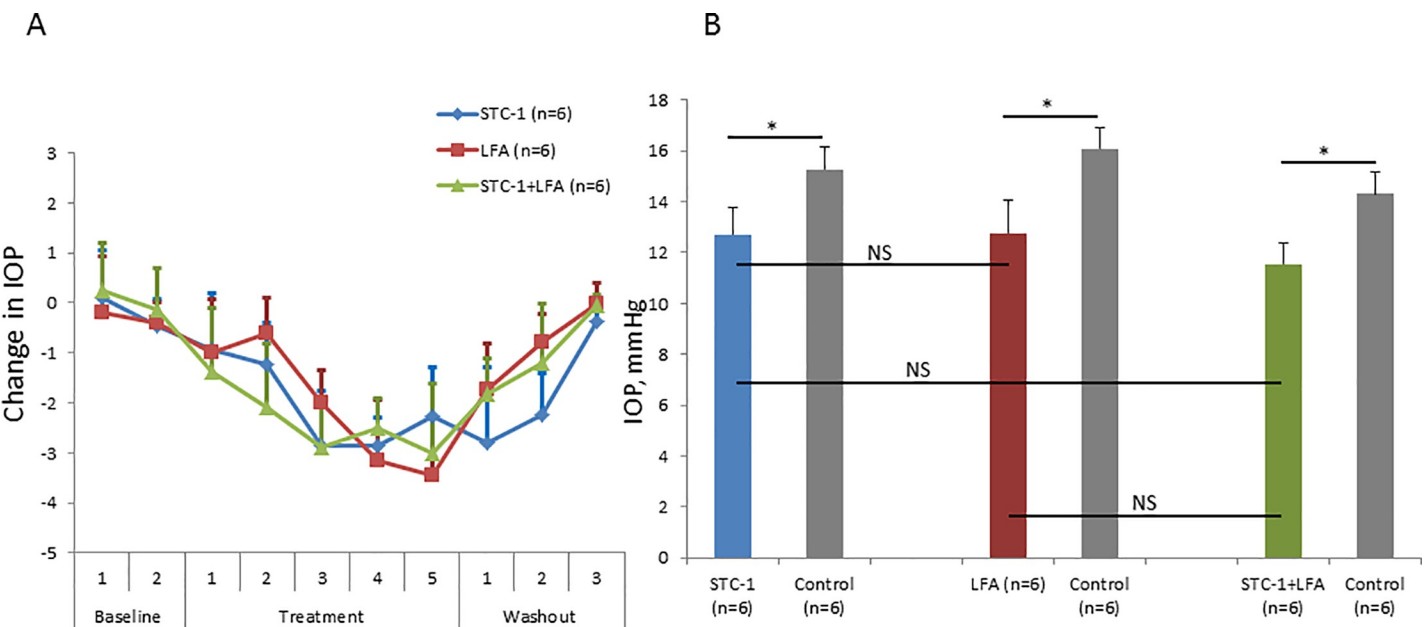

**Fig 1. STC-1 is equivalent and non-additive to LFA for IOP reduction in C57BL/6J wild-type mice.** A) After baseline IOP measurements, STC-1, LFA, and a combination of STC-1 and LFA all reduced IOP and returned to baseline following washout. B) At the maximal reduction of IOP with treatment (average of treatment day 4 and 5), LFA ($10^{-4}$ M), STC-1 (0.5 µg/µL) and LFA/STC-1 combination therapy in C57BL/6J mice all showed significant IOP reduction when compared to the contralateral vehicle control eye. However, no statistical difference in IOP reduction was seen when treatment groups were compared to one another. Sample size of all conditions was n = 6. NS = not significant. Error bars represent average ± standard deviation. * P<0.05.

by PCR showing the presence of the 35bp Schier stop cassette (Fig 3C). ELISA verified lack of FP receptor protein in FP receptor knockout mice and reduced FP receptor protein in mice heterozygous for the FP receptor (Fig 3D).

## Testing LFA and STC-1 in FP receptor knockout mice

We performed a longitudinal experiment where following baseline IOP measurements, C57BL/6J wild-type and FP receptor knockout mice were treated initially with LFA and then with STC-1 following a washout period. In this longitudinal experiment, LFA reduced IOP in wild-type but not FP receptor knockout mice while STC-1 reduced IOP in both wild-type and FP receptor knockout mice (Fig 4A). At treatment days 4–6, both LFA (14.7 ± 0.5 vs. 17.3 ± 0.6 mmHg, 18% reduction, P<0.05, n = 7) and STC-1 (15.0 ± 0.4 vs 18.1 ± 0.6 mmHg, 21% decrease, P<0.05, n = 7) significantly reduced IOP in wild-type mice (Fig 4B). Comparison of IOP reduction between LFA and STC-1 treated wild-type mice showed no significant difference (P>0.2, Fig 4B). In contrast, treatment of FP receptor knockout mice with LFA showed no IOP reduction (16.9 ± 0.3 vs 16.9 ± 0.4 mmHg, 0.0% change, P>0.9, n = 7, Fig 4B), consistent with AL-8810 FP receptor antagonist experiments in C57BL/6J mice (see Fig 2). In contrast to LFA, addition of STC-1 to FP receptor knockout mice lowered IOP (14.2 ± 0.3 vs 17.5 ± 0.6 mmHg, 24% decrease, P<0.05, n = 7). Comparison of IOP reduction in LFA and STC-1 treated FP receptor knockout mice showed a significant difference (P<0.05, Fig 4B) confirming that STC-1 lowers IOP through a non-FP receptor-mediated signaling pathway. Finally, there was a different paired response to treatment depending on whether wild-type or FP receptor knockout mice were used (P<0.0001).

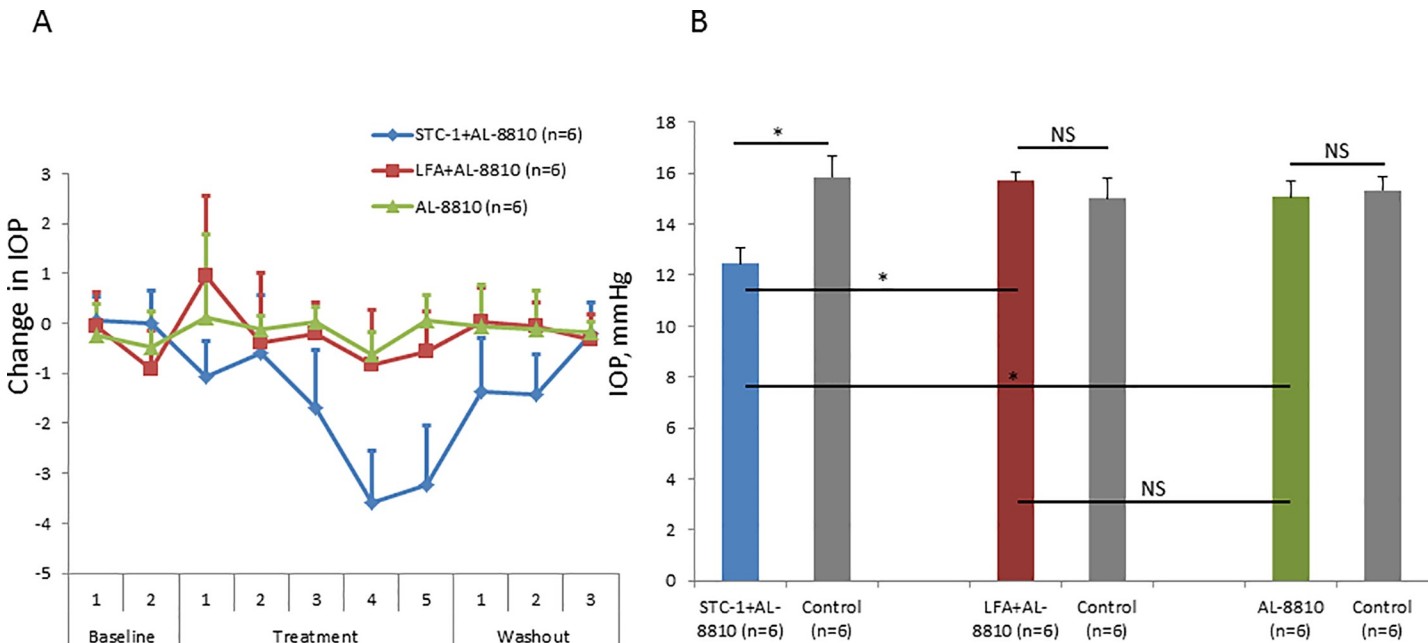

**Fig 2. STC-1, but not LFA, reduces IOP in the presence of a pharmacologic inhibitor of the FP receptor in wild-type mice.** After baseline IOP measurements, the pharmacologic FP receptor blocker AL-8810 (10 mM) was given alone or in combination with LFA ($10^{-4}$ M) or STC-1 (0.5 μg/μL). A) After pre-treatment baseline IOP measurements, STC-1 + AL-8810 reduced IOP; however, LFA + AL-8810 and AL-8810 alone had minimal effect on IOP. B) At the maximal reduction of IOP with treatment (average of treatment day 4 and 5), STC-1 + AL-8810 significantly reduced IOP compared to the contralateral eye. Neither LFA + AL-8810 or AL8810 alone affected pressure when compared to the contralateral eye. There was a significant IOP difference when comparing STC-1 + AL-8810 and LFA + AL-8810. There was no difference between LFA + AL-8810 and AL-8810 alone. Sample size of all conditions was n = 6. NS = not significant. Error bars represent average ± standard deviation. * P<0.05.

## Discussion

PGF2α analogues such as latanoprost are generally well-tolerated and effective in reducing elevated IOP in many patients with ocular hypertension and glaucoma. However, their use is limited in some patient populations due to side effects or lack of treatment response [6–16]. Because of their restricted use, development of new therapeutics that can treat these patient populations would be welcomed. Previously, we reported that STC-1 may be a candidate molecule as it is necessary for latanoprost-mediated IOP reduction and when applied topically can reduce IOP by itself. In the current study, we determined that IOP reduction with STC-1 is equivalent, but not additive, to LFA in C57BL/6J wild-type mice, and that STC-1-mediated IOP reduction does not require the FP receptor.

The finding that STC-1 does not require the FP receptor for its IOP lowering affect may have significant advantages for patients currently being treated with PGF2α analogues. Although little is known about PGF2α signaling in the eye, studies in non-ocular tissues have shown that activation of the FP receptor is associated with pro-inflammatory and pathologic effects. Elevated levels of PGF2α have been observed in patients with diseases characterized by inflammation including rheumatologic disease, obesity, and diabetes [27]. Furthermore, inhibition of PGF2α by pharmacologic blockade of the FP receptor with competitive antagonist AL-8810 has been shown to reduce the inflammatory response[28] and be therapeutic in animal models of stroke[29], traumatic brain injury[30], and multiple sclerosis[31]. Because patients treated with PGF2α analogues may have pro-inflammatory-like side effects, it is conceivable that these occur by binding to the FP receptor and its sequential activation of a number of signaling pathways unrelated to IOP reduction Therefore, identification of a novel

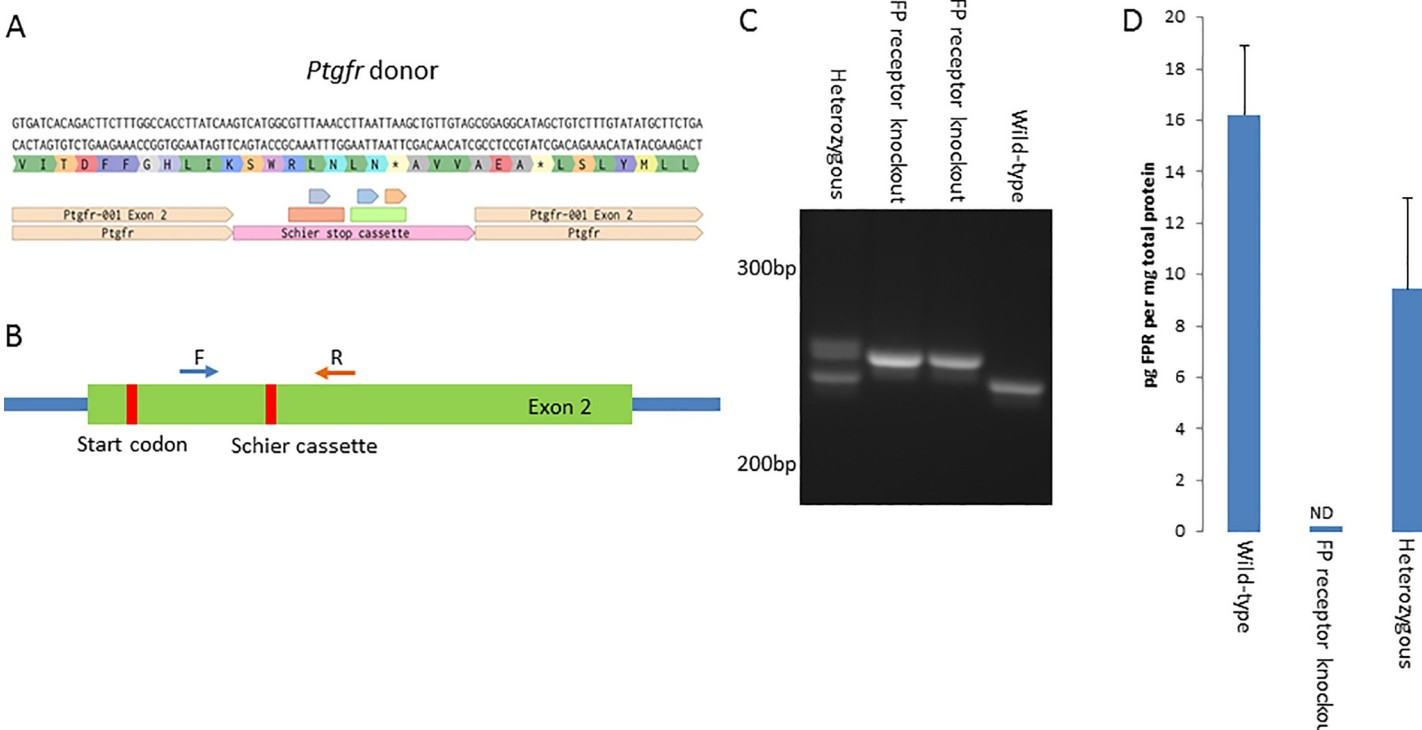

**Fig 3. Generation of FP receptor knockout mice.** A) Schematic representing strategy of Schier stop cassette insertion, single-guide RNA (sgRNA), WT Cas9 mRNA and a single-strand DNA (ssDNA) donor into the cytoplasm of C57BL/6NHsd zygotes. The ssDNA donors contained stop codons in all three reading frames to ensure a premature stop when inserted. B) Schematic representing Shier stop cassette. C) Genotyping data confirms insertion of the Schier stop cassette. D) ELISA in uterine tissue revealed diminished FP receptor expression at the protein level in mice heterozygous for the FP receptor compared to wild-type and non-detectible FP receptor expression in FP receptor knockout mice. ND = not detected. Error bars represent average ± standard deviation.

molecule downstream of PGF2α also has the potential to have a reduced side-effect profile for patients that have experienced conjunctival hyperemia, ocular surface irritation, hyper-pigmentation of the iris and periocular skin, orbital fat atrophy, hypertrichosis, [13] intraocular inflammation, [14, 15] reactivation of herpes simplex keratitis, or macular edema [16] with traditional PGF2α analogues. It remains to be seen whether the 20% of patients with reduced or no response to topical PGF2α that has been associated with SNPs in the FP receptor [7–12] may have benefit from topical STC-1. Additionally, it has been proposed that the poor IOP-lowering with latanoprost treatment observed in patients with Axenfeld-Rieger malformation is due to abnormal signaling in the FOXC1-FP receptor signaling axis [32, 33]. Therefore patients with variants in FP receptor signaling or the FP receptor itself may benefit from a therapy such as STC-1.

STC-1 is 50 kDa disulfide-linked dimer of two 25-kDa subunits that functions as a peptide hormone and is expressed in a variety of tissues including bone, skeletal muscle, heart, thymus and spleen [34, 35]. STC-1 was first described as a calcium regulatory protein in fish, being secreted from the corpuscles of Stannius to regulate calcium excretion at the gills and gut during hypercalcemia [35–37]. In addition to calcium metabolism, STC-1 has been shown to be cytoprotective by regulating oxidative stress [38–43] and inflammation, [43–46] and be neuroprotective for cerebral neurons, retinal photoreceptors, and retinal ganglion cells [47–50]. More recently, we demonstrated that STC-1 also has a role in development of the oxygen induced retinopathy stress response [51]. Though recent work has shown that megalin can act as a shuttle protein to transport STC-1 to the mitochondria, a potential cellular receptor has

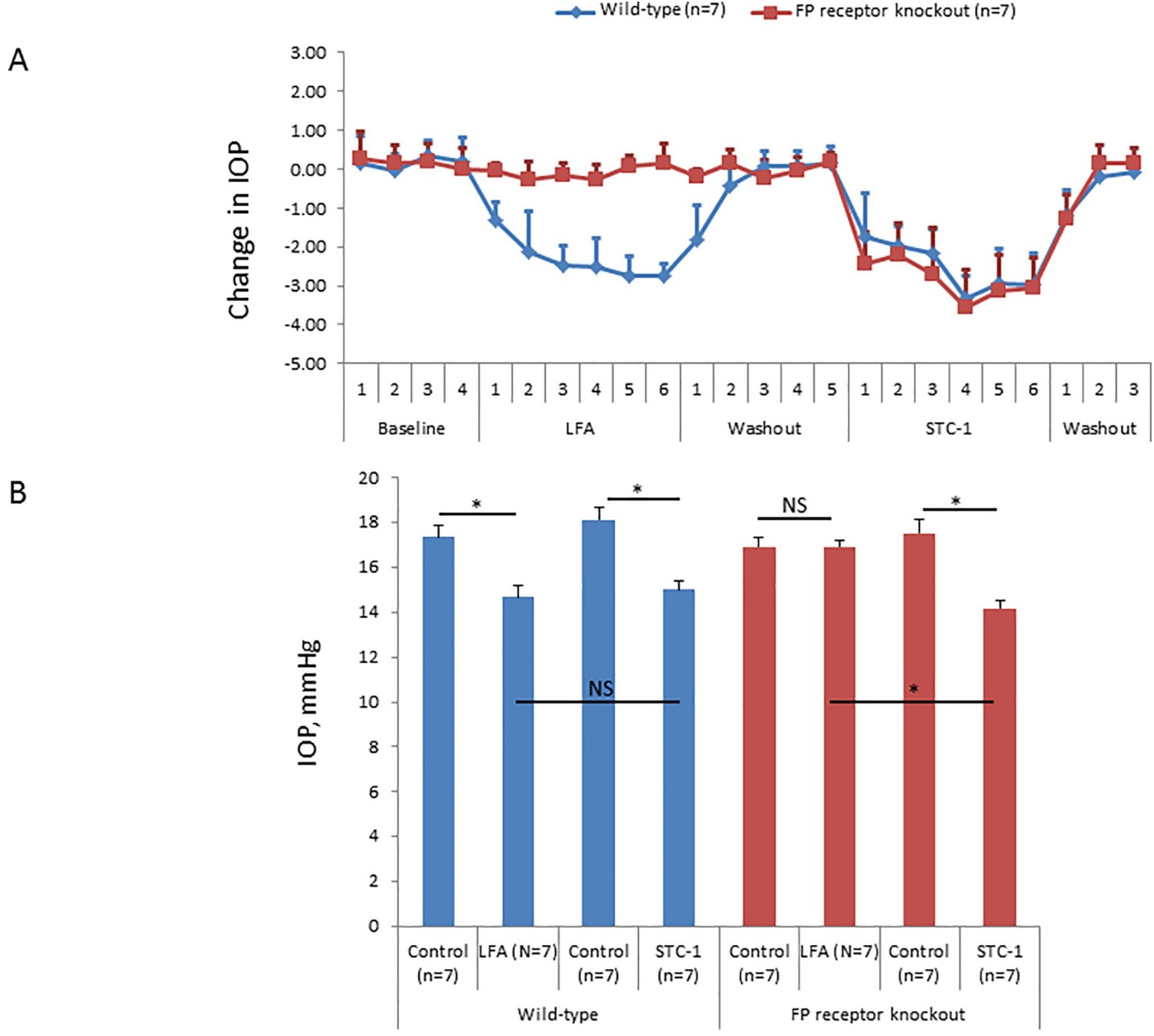

**Fig 4. STC-1, but not LFA, reduces IOP in the FP receptor knockout mouse.** After baseline IOP measurements, LFA ($10^{-4}$ M) was applied topically to both C57BL/6J wild-type and FP receptor knockout mice. Following treatment cessation and IOP returning to baseline, mice were treated with STC-1 (0.5 μg/μL). A) LFA reduced IOP in wild-type but not FP receptor knockout mice. STC-1 reduced IOP in both wild-type and FP receptor knockout mice. B) At the maximal reduction of IOP with treatment (average of treatment days 4–6) in wild-type mice, LFA and STC-1 each significantly reduced IOP compared to the contralateral control eye. There was no significant difference when comparing the LFA and STC-1 treated eyes. In FP receptor knockout mice, LFA did not alter IOP; however, STC-1 significantly reduced IOP compared to the vehicle control eye. There was a significant difference when comparing LFA and STC-1 treatment of FP receptor knockout mice. There was a significant difference to the treatments depending on whether wild-type or FP receptor knockout mice were used. NS = not significant. Error bars represent average ± standard deviation. * P<0.05.

not been identified [52]. Additional work defining the mechanism of action for STC-1 in IOP reduction is still needed including identification of a specific cellular receptor.

It is not clear whether STC-1 is replicating the actions of latanoprost by acting in a redundant pathway or in a separate, parallel pathway. Our findings that $10^{-4}$ M LFA and 0.5 μg/μL STC-1 reduced IOP to a similar amount ex vivo [23, 53] and in vivo do not provide clarification to this question. Additionally, our findings that LFA does not reduce IOP in the STC-1 knockout mouse and that STC-1 does reduce IOP in the FP receptor knockout mouse only suggests that there may be different signaling events that occur in the cascade between addition of LFA and upregulation of STC-1 and potentially common signaling events that occur whether STC-1 is upregulated following LFA addition or addition of STC-1 recombinant protein that ultimately leads to IOP reduction. The specific effects on outflow pathways are also not clear at this time. The human anterior segment culture model is thought to be primarily a model of trabecular outflow [54] yet both latanoprost [53] and STC-1 [23] have increased outflow facility in this model system. Latanoprost is classically thought to be a drug of uveoscleral outflow in people [13] while the distinct effects on uveoscleral and trabecular outflow in mice are not known [55]. Finally, our current and prior study [23] showing the IOP lowering effects of STC-1 used normotensive mice. Future studies with STC-1 will address whether it has the same IOP lowering properties in ocular hypertensive animal models. These studies will enable the assessment of STC-1 treatment on aqueous humor dynamics and its impact on trabecular meshwork function.

In summary, we have identified STC-1, a downstream effector molecule in latanoprost signaling, to be equivalent and non-additive in the early IOP response in wild-type mice. Additionally, we have shown that unlike latanoprost, STC-1 does not use the FP receptor for IOP reduction which may have important implications for patients who are unable to use standard PGF2α analogues due to pro-inflammatory side effects or minimal therapeutic response.

## Supporting information

**S1 Fig. Unaltered genotyping PCR in Fig 3C.**
(TIF)

## Author Contributions

**Conceptualization:** Gavin W. Roddy.

**Data curation:** Gavin W. Roddy, Tommy A. Rinkoski, Kjersten J. Monson, Uttio Roy Chowdhury.

**Formal analysis:** Gavin W. Roddy.

**Funding acquisition:** Gavin W. Roddy, Michael P. Fautsch.

**Investigation:** Gavin W. Roddy.

**Methodology:** Gavin W. Roddy.

**Resources:** Gavin W. Roddy.

**Supervision:** Michael P. Fautsch.

**Validation:** Gavin W. Roddy.

**Writing – original draft:** Gavin W. Roddy.

**Writing – review & editing:** Gavin W. Roddy, Michael P. Fautsch.

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
