## [Decision Letter · Decision Letter 0]

17 Mar 2020

PONE-D-20-03925

Stanniocalcin-1 (STC-1), a downstream effector molecule in latanoprost signaling, acts independent of the FP receptor for intraocular pressure reduction

PLOS ONE

Dear Dr. Roddy,

Thank you for submitting your manuscript to PLOS ONE. After careful consideration, we feel that it has merit but does not fully meet PLOS ONE’s publication criteria as it currently stands. Therefore, we invite you to submit a revised version of the manuscript that addresses the points raised during the review process.

ACADEMIC EDITOR: 

Please carefully address the reviewer concerns, which are clearly detailed.

We would appreciate receiving your revised manuscript by May 01 2020 11:59PM. To enhance the reproducibility of your results, we recommend that if applicable you deposit your laboratory protocols in protocols.io, where a protocol can be assigned its own identifier (DOI) such that it can be cited independently in the future. For instructions see: http://journals.plos.org/plosone/s/submission-guidelines#loc-laboratory-protocols

We look forward to receiving your revised manuscript.

Kind regards,

Ted S Acott, PhD

Academic Editor

PLOS ONE

Additional Editor Comments (if provided):

Reviewer concerns are clearly detailed and should be carefully addressed.

Journal Requirements:

"Research reported in this publication was supported by the National Eye Institute grants EY21727 (MPF) and EY26490 (MPF), Mayo Foundation (GWR, MPF), American Glaucoma Society (GWR), American Society of Cataract and Refractive Surgery (GWR), and Heed Foundation (GWR)"

3. We note that you have a patent relating to material pertinent to this article. Please provide an amended statement of Competing Interests to declare this patent (with details including name and number), along with any other relevant declarations relating to employment, consultancy, patents, products in development or modified products etc. Please confirm that this does not alter your adherence to all PLOS ONE policies on sharing data and materials, as detailed online in our guide for authors http://journals.plos.org/plosone/s/competing-interests by including the following statement: "This does not alter our adherence to  PLOS ONE policies on sharing data and materials.” If there are restrictions on sharing of data and/or materials, please state these. Please note that we cannot proceed with consideration of your article until this information has been declared.

Reviewers' comments:

Reviewer's Responses to Questions

**Comments to the Author**

1. Is the manuscript technically sound, and do the data support the conclusions?

Reviewer #1: Yes

Reviewer #2: Yes

2. Has the statistical analysis been performed appropriately and rigorously? 

Reviewer #1: Yes

Reviewer #2: No

3. Have the authors made all data underlying the findings in their manuscript fully available?

Reviewer #1: Yes

Reviewer #2: Yes

4. Is the manuscript presented in an intelligible fashion and written in standard English?

Reviewer #1: Yes

Reviewer #2: Yes

5. Review Comments to the Author

Reviewer #1: In this manuscript, authors studied whether STC-1, a downstream effector molecule in latanoprost signaling lowers IOP wild-type mice compared to to latonoprost. Moreover, authors examined whether latanoprost and STC-1 require the FP receptor for IOP reduction. Major findings are ; STC-1 is

equivalent in IOP reduction comppared to latonoprost and non-additive in the early IOP response in wild-type mice. Additionally, unlike latanoprost, STC-1 does not use the FP receptor for IOP reduction.

This is solid study and important in the field glaucoma. Authors need to just clarify the following minor comments.

1) Figure 4 is misplaced. It appears that figures are not in sequence. Figure 4 is kept first.. this needs to be corrected.

2)it is unclear whether IOPs are measured in masked manner. Since IOP differences are small, it is important to have masked measurements.

3) Based on discussion, STC-1 may be working on different pathways than latonoprost. However, this does not explain why STC-1 ko prevents latonoprost lowering IOPs (in previous manuscript). It seems more likely that STC1 may be working completely different pathway. Authors need to just clarify where crosstalk between STC-1 and latonoprost signaling is occurring.

4) It is not clear whether STC-1 has any effects on TM outflow. Considering previous studies that point out cytoprotective effects of STC-1 against mitochondrial stress, oxidative etc..it may have a protective role in TM dysfunction in glaucoma. Authors should clarify this as well.

5) Would STC1 lower IOP in similar fashion in ocular hypertensive or glaucoma mice? Please discuss this as well.

Reviewer #2: Manuscript ID: PONE-D-20-03925

Title: Stanniocalcin-1 (STC-1), a downstream effector molecule in latanoprost signaling, acts independent of the FP receptor for intraocular pressure reduction

Authors: Gavin W. Roddy, Tommy A Rinkoski, Kjersten J Monson, Uttio Roy Chowdhury, Michael P Fautsch.

Summary and Impression: Roddy et al present a paper surrounding Sanniocalcin-1 (STC-1), a molecule shown to be a downstream effector in latanoprost signaling. The authors suggested that use of the FP receptor for IOP lowering may be circumvented by using this molecule in lieu of PTGFR ligands such as latanoprost. To address this the authors, use a mouse model to show that STC-1 lowers IOP upon tonometry in comparable measures to latanoprost free acid at similar doses. They also show that the STC1 is able to lower IOP in a PTGFR knockout model in the same mouse background. This has implications for glaucoma treatments in cases where PGF2a analogues cannot be used due to genetic factors.

Overall the data is very interesting and obviously the authors have put a lot of work into this, especially in creation of the mouse model, and the data is believable. However, the presentation of the data, and some of the analysis should be redone. The manuscript itself could use some reorganization/restructuring. First off, the figures are out of order and have no explicit indication that Figure 1 is indeed Figure 1, which made reviewing this a little confusing. Additionally, the way the methods are laid out are confusing as is and I would request the authors rewrite parts of it for the sake of clarity. I also have concerns about some of the statistics. My major and minor issues are outlined below.

Major Issues:

1) I believe a Student paired T-test is inappropriate for the comparisons that are being done. As it stands now the authors do a series of paired tests between their groups of which N=6. The way the statistics are set up now lend itself to multiple testing errors (i.e. multiple pairwise comparisons). I suggest that the authors redo their stats with a Two-way ANOVA with an appropriate post-hoc test (A Bonferroni posthoc would be best for small data sets) rather than the series of paired tests. The authors are trying to demonstrate differences between treatments-controls and between treatments, thus a two-way ANOVA would be appropriate. This should take care of the below:

A) In Figure 1B, the authors should do statistical tests between the treatment groups as it appears there is a lower IOP in the double treatment group, which may indicate an effect on the contralateral eye. Otherwise it should be shown that there Is no statistical difference between the two.

B) In Figure 4B similarly the authors should show statistical differences (or not) between the wildtype and FP knockout groups.

2) Portions of the manuscript have discussion and results blended together. The results section of a scientific manuscript should be for simply reporting the data, whereas interpretation or rationale should be provided in the discussion. For example, under the heading “Determining the effect of FP receptor inhibition on IOP….” Just report the data you obtained. The rationale for the design and the interpretation should be moved to discussion. This is also true in the figure legends, please remove rationale and interpretation from the figures. Additionally, in figure legends, please list statistical test, and expression of error calculations (i.e standard error or std deviation?)

3) Could the authors comment on why uterine tissue was used for the FP receptor quantification. The tests performed concern eye tissue (whole eye for example) which would be the proper control post-mortem.

4) The discussion is not very robust. This should be used to place your data in the context of the literature which I feel the authors have not done here. One aspect of this study that I believe the authors aren’t haven't fleshed out well enough is that they have identified a molecule which may circumvent the FP receptor. They point out in passing that there are people with SNPs in the PTGFR gene that have reduced responses. Dr. Michael Walter’s group (I will freely admit that some of this is work I had a hand in) has published work on so-called ‘recalcitrant glaucoma’ in Axenfeld-Rieger syndrome. Strungaru et al (PMID: 17197537) showed that ARS patients with POAG do not respond to IOP lowering medications (including prostaglandin medications). Doucette et al. (PMID: 29847662) showed that FOXC1 controls expression of the FP receptor. Ultimately, my suggestion is to rewrite this part of the discussion to drive home this part of the data. There is real world applicability here. I’m not asking the authors to cite the above, but thought it would provide some additional context to the author’s dataset.

Minor Issues:

1) Introduction: “G-coupled protein receptor”. Should be G-protein coupled receptor

2) Introduction: “Up to 20% of patients are either minimally response of non-responsive” Could the authors expand on this point a little as I think it would add some implications to the data the authors present. i.e. What type of patients don’t respond or is this consistent across subtypes of glaucoma? Primary open angle? Anterior segment dysgeneses? Closed angle?

3) Introduction: “pigmentation of the iris”. Most likely the authors are referring to hyper-pigmentation or darkening of the iris here. Make this clear

4) Methods: “Animal Experiments” very minor, but mention that you’re using mice here. “Animals” is all that’s used to described the procedures here.

5) Perhaps this is vernacular I’m unfamiliar with, but the uses of the term “treatment nadir” is not something I’ve come across. Could the authors clarify what they mean by this in the text? I see its clarified in the legend of Figure 1, but this needs to either be changed or stated earlier.

6) “Testing LFA and STC-1 in FP receptor knockout mice” – “However, addition of STC-1 to FP receptor (-/-) lowered IOP….”. This seems to be an unfinished sentence.

7) When referring to genes, convention is to italicize

8) Please clarify between cessation and washout periods. As one sounds like stopping treatment whereas the other sounds like a physical washing of the eyes.

9) What is the vehicle for these drugs?

10) I urge the authors to consider using * and/or ** for statistical significance, rather than listing the p-values below 0.00001. This makes the figures much less cumbersome

11) Figure 3C there are no error bars or statistical analysis. While the effect is clearly obvious this needs to be done

6. PLOS authors have the option to publish the peer review history of their article (what does this mean?). If published, this will include your full peer review and any attached files.

Reviewer #1: Yes: Gulab Zode

Reviewer #2: Yes: Lance P Doucette

---

## [Author Response · Author response to Decision Letter 0]

27 Mar 2020

Reviewer #1: In this manuscript, authors studied whether STC-1, a downstream effector molecule in latanoprost signaling lowers IOP wild-type mice compared to to latonoprost. Moreover, authors examined whether latanoprost and STC-1 require the FP receptor for IOP reduction. Major findings are ; STC-1 is equivalent in IOP reduction comppared to latonoprost and non-additive in the early IOP response in wild-type mice. Additionally, unlike latanoprost, STC-1 does not use the FP receptor for IOP reduction. This is solid study and important in the field glaucoma. Authors need to just clarify the following minor comments.

1) Figure 4 is misplaced. It appears that figures are not in sequence. Figure 4 is kept first this needs to be corrected. We clarified the text surrounding figure 4, by adding “In a longitudinal experiment” in line 169 so the sentence now reads “In a longitudinal experiment, LFA reduced IOP in wild type but not FP receptor knockout mice whileSTC-1 reduced IOP in both wild type and FP receptor knockout mice (Fig 4A)”. We additionally added “Fig 4b” in line 174 so the sentence now reads “Comparison of IOP reduction between LFA and STC-1 treated wild-type mice showed no significant difference (P>0.2, Fig 4B).” 

2) It is unclear whether IOPs are measured in masked manner. Since IOP differences are small, it is important to have masked measurements. For IOP experiments, we added to the methods section “For each experiment, a second laboratory member who was masked to treatment groups checked IOP at multiple points to ensure accurate data collection” in lines 102-103. 

3) Based on discussion, STC-1 may be working on different pathways than latonoprost. However, this does not explain why STC-1 ko prevents latonoprost lowering IOPs (in previous manuscript). It seems more likely that STC1 may be working completely different pathway. Authors need to just clarify where crosstalk between STC-1 and latonoprost signaling is occurring. Thank you for these comments. We added to the discussion the following “It is not clear whether STC-1 is replicating the actions of latanoprost by acting in a redundant pathway or in a separate, parallel pathway. Our findings that 10-4 M LFA and 0.5 μg/μL STC-1 reduced IOP to a similar amount ex vivo[24, 54] and in vivo do not provide clarification to this question. Additionally, our findings that LFA does not reduce IOP in the STC-1 knockout mouse and that STC-1 does reduce IOP in the FP receptor knockout mouse only suggests that there may be different signaling events that occur in the cascade between addition of LFA and upregulation of STC-1 and potentially common signaling events that occur whether STC-1 is upregulated following LFA addition or addition of recombinant protein that ultimately lead to IOP reduction” in lines 222-229.

4) It is not clear whether STC-1 has any effects on TM outflow. Considering previous studies that point out cytoprotective effects of STC-1 against mitochondrial stress, oxidative etc..it may have a protective role in TM dysfunction in glaucoma. Authors should clarify this as well. We added in the discussion the following regarding trabecular vs uveoscleral outflow: “The human anterior segment culture model is thought to be primarily a model of trabecular outflow [55] yet both latanoprost [54] and STC-1 [24] have increased outflow facility in this model system. Latanoprost is classically thought to be a drug of uveoscleral outflow in people [14] while the distinct effects on uveoscleral and trabecular outflow in mice are not known [56] in lines 230-233”

5) Would STC1 lower IOP in similar fashion in ocular hypertensive or glaucoma mice? Please discuss this as well. This is an important question that we currently do not have an answer for. As the reviewer is aware, animal models that depict POAG are lacking. Most of the models are fairly acute, and really represent ocular hypertension. In future studies, we are planning to examine the effect of STC-1 in a steroid induced model of ocular hypertension as well as in TGF-beta overexpression model. To mention this idea in this manuscript, we added the following sentences: “Finally, our current and prior study [24]showing the IOP –lowering effects of STC-1 used normotensive mice. Future studies with STC-1 will address whether it has the same IOP lowering properties in ocular hypertensive animal models. These studies will enable the assessment of STC-1 treatment on aqueous humor dynamics and its impact on trabecular meshwork function” in lines 233-237.

Reviewer #2: Manuscript ID: PONE-D-20-03925

Title: Stanniocalcin-1 (STC-1), a downstream effector molecule in latanoprost signaling, acts independent of the FP receptor for intraocular pressure reduction

Authors: Gavin W. Roddy, Tommy A Rinkoski, Kjersten J Monson, Uttio Roy Chowdhury, Michael P Fautsch.

Summary and Impression: Roddy et al present a paper surrounding Sanniocalcin-1 (STC-1), a molecule shown to be a downstream effector in latanoprost signaling. The authors suggested that use of the FP receptor for IOP lowering may be circumvented by using this molecule in lieu of PTGFR ligands such as latanoprost. To address this the authors, use a mouse model to show that STC-1 lowers IOP upon tonometry in comparable measures to latanoprost free acid at similar doses. They also show that the STC1 is able to lower IOP in a PTGFR knockout model in the same mouse background. This has implications for glaucoma treatments in cases where PGF2a analogues cannot be used due to genetic factors.

Overall the data is very interesting and obviously the authors have put a lot of work into this, especially in creation of the mouse model, and the data is believable. However, the presentation of the data, and some of the analysis should be redone. The manuscript itself could use some reorganization/restructuring. First off, the figures are out of order and have no explicit indication that Figure 1 is indeed Figure 1, which made reviewing this a little confusing. Additionally, the way the methods are laid out are confusing as is and I would request the authors rewrite parts of it for the sake of clarity. I also have concerns about some of the statistics. My major and minor issues are outlined below.

We confirmed figure order, re-arranged methods by separating the in vivo experiments into separate paragraphs, and re-did the statistics per the below discussion and response.

Major Issues:

1) I believe a Student paired T-test is inappropriate for the comparisons that are being done. As it stands now the authors do a series of paired tests between their groups of which N=6. The way the statistics are set up now lend itself to multiple testing errors (i.e. multiple pairwise comparisons). I suggest that the authors redo their stats with a Two-way ANOVA with an appropriate post-hoc test (A Bonferroni posthoc would be best for small data sets) rather than the series of paired tests. The authors are trying to demonstrate differences between treatments-controls and between treatments, thus a two-way ANOVA would be appropriate. This should take care of the below:

A) In Figure 1B, the authors should do statistical tests between the treatment groups as it appears there is a lower IOP in the double treatment group, which may indicate an effect on the contralateral eye. Otherwise it should be shown that there Is no statistical difference between the two.

B) In Figure 4B similarly the authors should show statistical differences (or not) between the wildtype and FP knockout groups.

We have re-done the statistical analysis. The statistics section in Methods now reads “Average IOP was calculated at the maximal reduction of IOP with treatment. In experiments in which there were 5 treatment days, day 4 and 5 were averaged and taken as a single value.For experiments in which there were 6 treatment days, days 4-6 were averaged and taken as a single value. Given the sample size and distribution, a Wilcoxon signed rank test was used to compare control versus treatment in the same animal. In comparisons of different conditions among animals, a Kruskal-Wallis test was used with a Bonferroni correction. To determine whether there was a different response to wild type versus FP receptor knockout mice, an analysis of variance (ANOVA) on the paired difference was used. Statistical significance was determined if P<0.05” in lines 126-133.

2) Portions of the manuscript have discussion and results blended together. The results section of a scientific manuscript should be for simply reporting the data, whereas interpretation or rationale should be provided in the discussion. For example, under the heading “Determining the effect of FP receptor inhibition on IOP….” Just report the data you obtained. The rationale for the design and the interpretation should be moved to discussion. This is also true in the figure legends, please remove rationale and interpretation from the figures. Additionally, in figure legends, please list statistical test, and expression of error calculations (i.e standard error or std deviation?). We have removed the text as recommended above and instead added “To determine whether the FP receptor was necessary for STC-1-mediated IOP reduction…” in line 149. 

3) Could the authors comment on why uterine tissue was used for the FP receptor quantification. The tests performed concern eye tissue (whole eye for example) which would be the proper control post-mortem. Uterine tissue was used due to high tissue expression of the FP receptor. We added this to the methods with the statement “Uterine tissue was selected due to high tissue expression of the FP receptor [27]…” in lines 92-93. 

4) The discussion is not very robust. This should be used to place your data in the context of the literature which I feel the authors have not done here. One aspect of this study that I believe the authors aren’t haven't fleshed out well enough is that they have identified a molecule which may circumvent the FP receptor. They point out in passing that there are people with SNPs in the PTGFR gene that have reduced responses. Dr. Michael Walter’s group (I will freely admit that some of this is work I had a hand in) has published work on so-called ‘recalcitrant glaucoma’ in Axenfeld-Rieger syndrome. Strungaru et al (PMID: 17197537) showed that ARS patients with POAG do not respond to IOP lowering medications (including prostaglandin medications). Doucette et al. (PMID: 29847662) showed that FOXC1 controls expression of the FP receptor. Ultimately, my suggestion is to rewrite this part of the discussion to drive home this part of the data. There is real world applicability here. I’m not asking the authors to cite the above, but thought it would provide some additional context to the author’s dataset. We thank the reviewer for their comments. We agree that this may be an important point that needs further discussion. We have added the following text: “Therefore, identification of a novel molecule downstream of PGF2α also has the potential to have a reduced side-effect profile for patients that have experienced conjunctival hyperemia, ocular surface irritation, hyper-pigmentation of the iris and periocular skin, orbital fat atrophy, hypertrichosis, [14] intraocular inflammation,[15, 16] reactivation of herpes simplex keratitis, or macular edema [17] with traditional PGF2α analogues. It remains to be seen whether the 20% of patients with reduced or no response to topical PGF2α that has been associated with SNPs in the FP receptor [8-13] may have benefit from topical STC-1. Additionally, it has been proposed that the poor IOP-lowering with latanoprost treatment observed in patients Axenfeld-Rieger Malformation is due to abnormal signaling in the FOXC1-FP receptor signaling axis [33, 34]. Therefore patients with variants in FP receptor signaling or the FP receptor itself may particularly benefit from a therapy such as STC-1” in lines 200-210. Additionally the discussion was further strengthened by addressing comments by request of reviewer 1, additional text was also added to the Discussion in lines 222-237.

Minor Issues:

1) Introduction: “G-coupled protein receptor”. Should be G-protein coupled receptor. Thank you for catching this. This was corrected in line 42.

2) Introduction: “Up to 20% of patients are either minimally response of non-responsive” Could the authors expand on this point a little as I think it would add some implications to the data the authors present. i.e. What type of patients don’t respond or is this consistent across subtypes of glaucoma? Primary open angle? Anterior segment dysgeneses? Closed angle?We have added the following text: “Up to 20% of patients with ocular hypertension or multiple types of glaucoma including low tension, primary open angle, exfoliative, or pigment dispersion have a reduced or absent responsePGF2α analogues[5-7]” in lines 45-47. 

3) Introduction: “pigmentation of the iris”. Most likely the authors are referring to hyper-pigmentation or darkening of the iris here. Make this clear. Thank you for catching this. This was corrected in line 51. 

4) Methods: “Animal Experiments” very minor, but mention that you’re using mice here. “Animals” is all that’s used to described the procedures here. This was changed to “Mouse Experiments” in line 67.

5) Perhaps this is vernacular I’m unfamiliar with, but the uses of the term “treatment nadir” is not something I’ve come across. Could the authors clarify what they mean by this in the text? I see its clarified in the legend of Figure 1, but this needs to either be changed or stated earlier. Treatment nadir was changed to “maximal reduction of IOP with treatment” throughout the manuscript for clarity. 

6) “Testing LFA and STC-1 in FP receptor knockout mice” – “However, addition of STC-1 to FP receptor (-/-) lowered IOP….”. This seems to be an unfinished sentence. For clarification, we changed “However” to “In contrast to LFA” in line 175. 

7) When referring to genes, convention is to italicize. We have now italicized gene names throughout the manuscript. 

8) Please clarify between cessation and washout periods. As one sounds like stopping treatment whereas the other sounds like a physical washing of the eyes. For clarification, after washout, we added “in which the animals received no treatment” in lines 111-112.

9) What is the vehicle for these drugs? We added the appropriate vehicle control. The sentence now reads: “In all cases, the contralateral eye was treated with vehicle (PBS for STC-1; dilution of dimethylsulfoxide (DMSO) 1:1000 in PBS for LFA, dilution of DMSO 1:2.5 in PBS for AL-8810)” in lines 108-110. 

10) I urge the authors to consider using * and/or ** for statistical significance, rather than listing the p-values below 0.00001. This makes the figures much less cumbersome. We have added asterisks to the figures and removed numbers of significance. 

11) Figure 3C there are no error bars or statistical analysis. While the effect is clearly obvious this needs to be done. We have added error bars to this graph.

---

## [Decision Letter · Decision Letter 1]

20 Apr 2020

Stanniocalcin-1 (STC-1), a downstream effector molecule in latanoprost signaling, acts independent of the FP receptor for intraocular pressure reduction

PONE-D-20-03925R1

Dear Dr. Roddy,

We are pleased to inform you that your manuscript has been judged scientifically suitable for publication and will be formally accepted for publication once it complies with all outstanding technical requirements.

With kind regards,

Ted S Acott, PhD

Academic Editor

PLOS ONE

Additional Editor Comments (optional):

concerns addressed

Reviewers' comments:

Reviewer's Responses to Questions

**Comments to the Author**

1. If the authors have adequately addressed your comments raised in a previous round of review and you feel that this manuscript is now acceptable for publication, you may indicate that here to bypass the “Comments to the Author” section, enter your conflict of interest statement in the “Confidential to Editor” section, and submit your "Accept" recommendation.

Reviewer #1: All comments have been addressed

Reviewer #2: All comments have been addressed

2. Is the manuscript technically sound, and do the data support the conclusions?

Reviewer #1: Yes

Reviewer #2: Yes

3. Has the statistical analysis been performed appropriately and rigorously? 

Reviewer #1: Yes

Reviewer #2: Yes

4. Have the authors made all data underlying the findings in their manuscript fully available?

Reviewer #1: Yes

Reviewer #2: Yes

5. Is the manuscript presented in an intelligible fashion and written in standard English?

Reviewer #1: No

Reviewer #2: Yes

6. Review Comments to the Author

Reviewer #1: (No Response)

Reviewer #2: I believe the authors have addressed the reviewer's comments sufficiently. I thank the editors for the opportunity to review this paper, and wish the authors the best of luck in their future endeavors.

7. PLOS authors have the option to publish the peer review history of their article (what does this mean?). If published, this will include your full peer review and any attached files.

Reviewer #1: Yes: GulabZode

Reviewer #2: Yes: Lance P Doucette

---

## [Editor Report · Acceptance letter]

24 Apr 2020

PONE-D-20-03925R1 

Stanniocalcin-1 (STC-1), a downstream effector molecule in latanoprost signaling, acts independent of the FP receptor for intraocular pressure reduction 

Dear Dr. Roddy:

I am pleased to inform you that your manuscript has been deemed suitable for publication in PLOS ONE. Congratulations! Your manuscript is now with our production department. 

With kind regards,

on behalf of

Dr. Ted S Acott 

Academic Editor

PLOS ONE